# Antimicrobial Peptide with a Bent Helix Motif Identified in Parasitic Flatworm *Mesocestoides corti*

**DOI:** 10.3390/ijms252111690

**Published:** 2024-10-30

**Authors:** Tomislav Rončević, Marco Gerdol, Sabrina Pacor, Ana Cvitanović, Anamarija Begić, Ivana Weber, Lucija Krce, Andrea Caporale, Mario Mardirossian, Alessandro Tossi, Larisa Zoranić

**Affiliations:** 1Department of Biology, Faculty of Science, University of Split, 21000 Split, Croatia; troncevic@pmfst.hr; 2Department of Life Sciences, University of Trieste, 34127 Trieste, Italy; mgerdol@units.it (M.G.); pacorsab@units.it (S.P.); mmardirossian@units.it (M.M.); atossi@units.it (A.T.); 3Department of Biology, Faculty of Science, University of Zagreb, 10000 Zagreb, Croatia; ana.cvitanovic@student.pmf.hr; 4Department of Physics, Faculty of Science, University of Split, 21000 Split, Croatia; abegic@pmfst.hr (A.B.); ivana@pmfst.hr (I.W.); lkrce@pmfst.hr (L.K.); 5Institute of Crystallography, CNR, Basovizza, 34149 Trieste, Italy; andrea.caporale@cnr.it

**Keywords:** antimicrobial peptides, membranolytic activity, bent helix conformation

## Abstract

The urgent need for antibiotic alternatives has driven the search for antimicrobial peptides (AMPs) from many different sources, yet parasite-derived AMPs remain underexplored. In this study, three novel potential AMP precursors (mesco-1, -2 and -3) were identified in the parasitic flatworm *Mesocestoides corti*, via a genome-wide mining approach, and the most promising one, mesco-2, was synthesized and comprehensively characterized. It showed potent broad-spectrum antibacterial activity at submicromolar range against *E. coli* and *K. pneumoniae* and low micromolar activity against *A. baumannii*, *P. aeruginosa* and *S. aureus*. Mechanistic studies indicated a membrane-related mechanism of action, and circular dichroism spectroscopy confirmed that mesco-2 is unstructured in water but forms stable helical structures on contact with anionic model membranes, indicating strong interactions and helix stacking. It is, however, unaffected by neutral membranes, suggesting selective antimicrobial activity. Structure prediction combined with molecular dynamics simulations suggested that mesco-2 adopts an unusual bent helix conformation with the N-terminal sequence, when bound to anionic membranes, driven by a central GRGIGRG motif. This study highlights mesco-2 as a promising antibacterial agent and emphasizes the importance of structural motifs in modulating AMP function.

## 1. Introduction

Over the last few decades, infections caused by antibiotic-resistant bacteria have become a global problem and have been responsible for millions of deaths worldwide, estimated at 1.3 million in 2019 and contributing to 5 million deaths in total [1]. The majority of these infections are due to *Escherichia coli*, *Staphylococcus aureus*, *Klebsiella pneumoniae*, *Streptococcus pneumoniae*, *Acinetobacter baumannii* and *Pseudomonas aeruginosa*, shortlisted by the World Health Organization (WHO) as top-priority pathogens for which new drugs are urgently needed [2]. The available antibiotics are increasingly overwhelmed by the surge in these multidrug- and pandrug-resistant bacteria, but only a few novel drugs with truly novel mechanisms of action are currently being developed.

Antimicrobial peptides (AMPs) are endogenous molecules that are synthesized as part of the innate defenses of all multicellular organisms and have the potential to mitigate antimicrobial resistance. They act directly against bacteria, often with broad-spectrum activity against Gram-negative and Gram-positive microorganisms, and may also have immunomodulatory effects that enhance other aspects of immunity [3]. The most common mechanism of action involves the disruption of bacterial membranes in a relatively non-specific manner, which makes it difficult for target microorganisms to acquire lasting resistance, although non-permeabilizing modes of action have also been described [4].

Few AMPs, however, have reached late-stage clinical trials so far, as their antimicrobial potency is accompanied by low bioavailability and stability in vivo and unacceptable levels of toxicity towards host cells [5]. Different strategies, such as chemical modifications and the use of delivery vehicles, have been used to try and overcome these obstacles, but with limited success [6,7]. An alternative approach is to design artificial AMPs de novo with reduced toxicity while maintaining favorable antimicrobial properties. This requires the development of molecular descriptors that link structural and biophysical properties with favorable biological activity obtained from quantitative structure–activity relationship (QSAR) studies, which is not trivial but, when effective, can provide the means to design new drugs in a fast and cost-effective way [8].

A potentially useful strategy to identify naturally occurring peptides with useful properties is to combine omics technologies and bioinformatic analyses [9]. According to the CAMPR4 database, almost 12,000 natural AMPs have been identified so far, with the majority in animals [10]. Among these, the AMPs present in parasitic helminths (Nematoda and Platyhelminthes) are poorly represented, but we believe that they are particularly interesting as these animals have to adapt to different environments during their life cycles and coexist with their mammalian hosts, making them a potential source of safe, biologically active molecules [11,12]. We have successfully identified anisaxins, a cecropin-like family of peptides identified in the marine nematodes *Anisakis* spp., with potent activity against both Gram-negative and Gram-positive bacteria and little or no toxicity against tested animal cells [13]. The flatworm *Mesocestoides corti* is another parasitic organism that infects small animals (mice, birds, frogs, etc.) as intermediate hosts and then small carnivores (cats, dogs, foxes, etc.) as the final host and can incidentally infect humans [14]. We have therefore extended our attention to this parasite by implementing a genome-wide mining approach. We have successfully identified three possible AMP sequences. One of these peptides, termed mesco-2, was selected for extensive characterization, which included structural studies and the determination of its antimicrobial activity and toxicity. This was coupled with mode-of-action studies using fluorescent microscopy, atomic force microscopy (AFM) and flow cytometry, whereas findings at the atomistic level regarding peptide behavior were obtained with molecular dynamics (MD) simulations.

## 2. Results

### 2.1. Peptide Sequences

A genome-wide mining approach led to the identification of three protein precursors that fulfilled all the required criteria, encoded by the genes MCU_005415, MCU_006988 and MCU_000972. While the first two genes were complete and intronless, thereby displaying an uninterrupted open reading frame that facilitated the prediction of the encoded protein sequence, MCU_000972 was characterized by the presence of a short internal undetermined region. The full-length cDNA sequence of MCU_000972 was therefore reconstructed from de novo-assembled transcriptome data. The complete protein sequences were named mesco-1, -2 and -3. A fourth homologous sequence, encoded by an incomplete gene (MCU_006991), was placed at the terminus of a truncated genomic contig and was therefore not subjected to further analysis.

The expression of the three complete genes was investigated in the larval (TT) and strobilated worm (ST) developmental stages, identifying a variable transcriptional level that was consistently higher in ST, with a mean fold increase of 7.5× for mesco-2 and 8.5× for mesco-3 but <2× for mesco-1 (Appendix A). Considering the physicochemical properties, all three sequences would result in peptides with favorable characteristics (cationicity, amphipathicity, etc.). However, mesco-2, despite having the lowest hydrophobicity, was predicted to potentially adopt a helical conformation with the highest hydrophobic moment (see Table 1), which could correlate with a better potency/toxicity balance [5]. Mesco-2 was therefore selected for chemical synthesis and characterization. It consists of a cationic (+11), 40-aa-long peptide with an N-terminal stretch, predicted to adopt a helical conformation, with a likely disulfide-bridged segment located near the C-terminus. In some respects, this resembles the arrangement in some anuran AMPs that carry the so-called Rana-box domain [15], although the sequence of the linear domain and the sequence and spacing of the box domain are quite different.

### 2.2. Antibacterial Activity

The antibacterial activity of mesco-2 was assessed against a panel of Gram-negative bacteria, including *Escherichia coli*, *Klebsiella pneumoniae*, *Acinetobacter baumannii* and *Pseudomonas aeruginosa*, as well as two different strains of *Staphylococcus aureus* as representatives of Gram-positive bacteria. It proved to be quite potent and broad-spectrum, with minimal inhibitory concentrations (MIC) ranging from submicromolar (e.g., 0.5 μM for *E. coli* and *K. pneumoniae*) to low micromolar (1–2 μM for *A. baumannii*, *P. aeruginosa* and *S. aureus* depending on the strain).

The minimal bactericidal concentrations (MBC) against all tested strains were similar to the MIC values, suggesting that mesco-2 is bactericidal rather than bacteriostatic (Table 2), with one possible exception (the MBC for *S. aureus* ATCC 25923 was four times higher).

The antimicrobial activity was assessed also in terms of the bacterial growth inhibition of the *E. coli* ATCC 25922 and *S. aureus* ATCC 29213 strains, in the presence of increasing peptide concentrations (Figure 1). This is more sensitive than the MIC assay as it allows for the observation of peptide activity also at sub-lethal concentrations. Mesco-2, in fact, seemed to interact with bacteria and affect growth at concentrations well below the MIC, as, even at the lowest tested concentrations (0.03–0.06 μM), the growth of *E. coli* was prevented for 2–3 h compared to the control, after which the bacteria were able to compensate and start rapidly growing (Figure 1). At one-quarter of the MIC (0.0125 μM), the growth inhibition was prolonged to 6 h compared to the control, after which some bacterial growth could be observed for up to 12 h. There was no detectable bacterial growth at half of the MIC (0.25 μM) for the duration of the measurement.

*S. aureus* followed a similar pattern, and the highest peptide concentration for which the bacteria were eventually able to compensate for the presence of the peptide was 0.25 μM (one-quarter of the MIC) (Figure 1). After 12 h, the total number of bacteria was still considerably lower than that in the control group. Interestingly, at the concentrations at which the bacteria did compensate, the growth delay compared to the control was far smaller than for *E. coli*, at 1–2 h depending on the concentration (Figure 1).

### 2.3. Toxicity

MEC-1 cells represent a useful model to explore cytotoxic effects as they grow in suspension and provide a dynamic, functional and broad membrane surface for peptides in solution. The dose–response curves show the effects of mesco-2 on the viability and membrane integrity of MEC-1 cells (Figure 2). The viability (blue line) was assessed after 24 h incubation with the peptide at concentrations ranging from 0.1 to 100 µM in complete medium, and the same range of concentrations was used for the flow cytometric analysis of PI uptake to assess the membrane damage (red line) after 30 min of exposure to the peptide. The damaging effects of mesco-2 depend on a rather rapid membrane-permeabilizing effect (within 30 min), leading to the loss of cell viability with cytotoxic activity at concentrations higher than those required to kill the majority of the bacteria, thus showing some selectivity for bacterial pathogens (Table 2 and Figure 2).

### 2.4. Peptide–Membrane Interaction

Given the fact that helical AMPs most often act by permeabilizing bacterial membranes [17,18], we visualized the effect of mesco-2 on the surfaces of *E. coli* DH5α cells using atomic force microscopy. The cells were treated at 4× MIC, having previously confirmed that this was the same as for the ATCC 25922 strain (MIC = 0.5 μM). After treatment, the bacterial cells preserved their morphologies and no evident signs of surface alterations were seen compared to the control (Figure 3a and Appendix A). However, fluorescence imaging on the same cells showed a strong propidium iodide (PI) signal (cells stained red), indicating that mesco-2 creates pores or lesions as part of its mechanism of action (Figure 3b,d). This was further confirmed by measuring the percentage of PI-positive *E. coli* ATCC 25922 cells at this mesco-2 concentration by flow cytometry. The peptide proved to rapidly disrupt the inner bacterial membrane, with ˃90% of cells being PI-positive cells at 0.5 μM (MIC) after 15 min (Figure 3c). The same type of behavior has been reported for the bee venom-derived peptide melittin, which caused strong PI-induced signals on both Gram-negative and -positive bacterial cells while leaving the cell surface apparently intact [19,20] and causing strong molecular leakage from both zwitterionic and anionic lipids [21]. Similar behavior has also been previously described for anisaxins, a family of helical membranolytic peptides identified from the parasitic helminths *Anisakis* spp. [13].

### 2.5. Secondary Structure

The tendency of mesco-2 to adopt secondary structures and alter them on membrane interaction was evaluated by circular dichroism (CD) spectroscopy. We used different isotropic and anisotropic environments, including (i) an aqueous solution at a neutral pH (sodium phosphate buffer, SPB); (ii) different proportions of trifluoroethanol (TFE) (5–50%) in 10 mM SPB; (iii) sodium dodecyl sulphate (SDS) micelles in 10 mM SPB; and (iv) membrane-mimicking environments in the form of neutral and anionic small unilamellar vesicles (SUVs), as models for eukaryotic and bacterial membranes, respectively. As expected, the peptide was unstructured in the aqueous buffer but started to display some helical content in the presence of TFE, reaching ≈20% helix content at approximately 40% TFE. Helix formation was also observed in the presence of the SDS micelles, which represented a simple membrane-mimicking environment. This was, however, lesser compared to the helical structuring with larger proportions of TFE (Figure 4).

Helical structuring could also be observed in the presence of negatively charged liposomes, even at the lowest liposome concentrations, which was not unexpected given the strong cationic nature of the peptide. The structure, however, differed significantly from that of a canonical α-helix, as was observed in the presence of TFE, which could possibly be attributed to helix stacking in the peptide [22] and may support structure predictions, as described below. On the other hand, the peptide showed almost no alteration in the CD spectra in the presence of neutral liposomes, suggesting a lower capacity to interact with eukaryotic membranes or a much weaker type of interaction that does not markedly alter the peptide’s conformation (Figure 4).

### 2.6. Structural Assessment via Molecular Modeling

The structural predictions were carried out using various web-based systems, and the predicted structures varied somewhat, so two characteristic structures were selected as starting structures for the modeling of membrane interactions. The first model (*model1*), generated by ColabFold, had a low confidence score but was the only one to predict a disulfide bond between Cys36 and Cys39 in mesco-2. This model featured a predominantly straight N-terminal α-helix with an unstructured C-terminal region (Figure 5). The second model (*model2*) was obtained using the AlphaFold Protein Structure Database (AlphaFold DB), predicting a structure containing an unusual bent helix, with two α-helical segments connected by a rather tight bend between residues Gly13 and Ile16, followed by a coil at the C-terminal with a disulfide bond. Again, this had a low pLDDT score, indicating uncertainty and a possibly unstructured nature in isolation (Figure 5).

Following the results from the structural prediction, the stability of the mesco-2 peptide was investigated through two sets of simulation experiments: one in water and the other in the vicinity of an anionic membrane, for both initial configurations of *model1* and *model2*. Each set of experiments included two cases (case1 and case2) of 500 ns simulation time, which differed in their starting velocities.

The simulations in water showed that *model1* lost its helical structure and unfolded (Figure 5 and Appendix A), preserving the α-helical conformation for only part of the sequence in both simulation cases, as shown by the DSSP plots (Appendix A). On the other hand, the simulations confirmed that *model2* was relatively stable in water (Figure 5 and Appendix A), maintaining most of its helix–coil–helix structure in both the case1 and case2 simulations (Appendix A).

When placed near an anionic membrane, both models demonstrated rapid binding, indicating a strong affinity for the membrane, primarily through interactions with polar residues (see Figure 6 and Appendix A). Upon membrane binding, *model1* showed different behavior in the two simulation cases, with unfolding at the N-terminal region in case1 and the formation of a coil structure from Arg14 to Ile16, as shown by visual inspection and DSSP analysis (Appendix A), which resembled the bent helix shape predicted for *model2* by AlphaFold DB. The peptide penetrated the membrane beyond the polar lipid region, as evidenced by the density profiles (see Figure 7 and Appendix A). In the case2 simulation (Appendix A), the formation of the inner coil was also observed, but the two helices at the ends were less stable than in case1, as indicated by the DSSP plots (Appendix A). In this case, the peptide also entered deep into the membrane, but not as much as in case1 (Appendix A). Time-dependent plots of the distance from the membrane center for polar and hydrophobic residues showed that as the peptide penetrated deeper into the membrane, there was a shift in the position of the polar and hydrophobic residues, with the hydrophobic residues moving closer to the membrane center (Appendix A).

Interaction with the membrane for *model2* resulted in the mostly preserved bent helix structure (Figure 6 and Appendix A), with the widening of the coil–bend structure (Phe9 to Gly15) in the case2 simulation, as shown in the DSSP plots (Appendix A). In both cases, the peptide remained on the membrane surface, with most of the contact occurring between the membrane and the polar residues (Figure 7 and Appendix A). The simulations also suggest that the C-terminal region has less affinity for membrane binding, as it remained in the water during the simulation in case1.

## 3. Discussion

There is strong pressure to uncover compounds that could provide alternatives to the currently used antibiotics and assist in the fight against resistant pathogens. AMPs of different origin, due to their native function in the innate immune responses of animals to pathogens, have been the focus of research for several decades [5]. However, the number of reported AMPs from parasites is still scarce. Nevertheless, helminth AMPs may have particularly useful properties as parasitic organisms must adapt to various environments throughout their life cycles, coexisting with different animal hosts, and have the ability to modulate their immune responses. AMPs originating from parasites could offer promising leads for drug development.

In this study, the genome-wide mining approach identified three protein precursors, named mesco-1, mesco-2 and mesco-3, from the parasitic flatworm *M. corti*. The expression levels of the three genes were higher in the strobilated worm (ST) developmental stage compared to the larval stage (TT), with mesco-2 and mesco-3 showing significant upregulation during this transition. Mesco-2 was chosen for synthesis due to its favorable physicochemical properties and potential biological activity, being a 40-amino-acid peptide with two cysteine residues and a disulfide bridge at the C-terminus.

A variety of online servers predicted that mesco-2 had high potential for antimicrobial activity, along with low toxicity, supporting its functionality as an antimicrobial peptide (see Appendix A). Confirmation came from measurements of the biological activity, which showed that mesco-2 exhibited excellent and broad-spectrum antibacterial activity with sub-micromolar MICs (0.5 µM) for *E. coli* and *K. pneumoniae* and low micromolar activity against *A. baumannii*, *P. aeruginosa* and *S. aureus* (1–2 µM). The minimal bactericidal concentrations were similar to the MICs, indicating bactericidal activity. MEC-1 cells, used to assess mesco-2’s cytotoxic effects, showed rapid membrane permeabilization within 30 min, leading to a loss of viability, but this was at higher concentrations than those required to kill bacteria, demonstrating some selectivity towards bacterial pathogens. These results confirm the potential of mesco-2 as an antibacterial agent.

Biophysical measurements using atomic force microscopy, fluorescence imaging and flow cytometry indicate that mesco-2 has a membrane-related mechanism of action. Namely, while the AFM signals showed that the bacterial surface appeared unaffected when exposed to a peptide concentration of 4× MIC, strong propidium iodide fluorescence was observed in the cells, indicating that the membrane was compromised. Flow cytometry further confirmed the rapid PI uptake after the exposure of the bacterial cells to mesco-2 at the MIC. These results suggest that mesco-2 may act by forming transient pores or lesions that allow the rapid passage of small molecules through the membrane, while preserving the overall membrane integrity over time.

The relationship between the structure (or structure-related properties) and biological activity is well explored and documented for AMPs [5]. Here, we assessed the structuring of mesco-2 using CD spectroscopy in various environments, as well as by molecular modeling and simulations. The CD spectra showed that while the peptide was unstructured in an aqueous environment, it could adopt a helical conformation, at least over part of its sequence, in the presence of TFE. This preference for the formation of a helical structure in a hydrophobic environment was further confirmed by the presence of helical content in SDS, a simple membrane-mimicking environment. The CD spectra of mesco-2 in solution with neutral SUVs indicated an unstructured state similar to that in water, suggesting that the peptide did not interact strongly with the neutral membranes. In contrast, the CD spectra of mesco-2 in a solution with anionic SUVs, even at low SUV concentrations, showed a high level of structuring. These spectra were significantly different from those measured in the presence of TFE, suggesting that it consists of stacked helices [22]. This is consistent with the bent helix conformation suggested by AlphaFold DB for this peptide.

The structure prediction servers ColabFold and AlphaFold DB revealed two distinct structures for mesco-2: one with a long N-terminal α-helix and an unstructured C-terminal region and another with an N-terminal bent helix structure and an unstructured C-terminal region. The stability of the two predicted models was tested using molecular dynamics simulations in water and near anionic membranes, with moderate simulation times of 500 ns. *Model1* somewhat resembles the structure reported for LL-37 [23], albeit with a longer C-terminal unstructured region, and there is also tenuous sequence resemblance at both ends of the helical segment. However, given the higher charge and the presence of several Gly residues in this region, it is likely to behave more like RL-37, which is structured as a lone helix only on contact with biological membranes, rather than LL-37, which oligomerizes in solution and likely approaches the membrane as a helical bundle [24].

The simulation of a single mesco-2 peptide in water showed that the bent helix structure was more stable, while the single-helix model mostly unfolded at both ends of the initial α-helix. In interaction with the membrane, the peptide with the initial bent helix structure binds strongly with its polar residues but remains on the membrane surface, while the peptide initially with the long N-terminal helix forms a helix–bend–helix structure, similar to the other model, and penetrates deeper into the membrane. These simulations may represent two facets of the peptide’s behavior on membrane interaction, with the peptide initially adopting a more linear helical conformation on membrane interaction and then a bent helix conformation, as indicated by spectra in the presence of anionic SUVs by CD. This highlights the importance of the central GRGIGRG motif, which is crucial for the coil/bend formation.

Regarding the structural motifs, glycine frequently appears at helix termini due to its propensity to adopt dihedral angles that disrupt helical structures, favoring turns or coils [25]. The GRG sequence may act as a helix cap, stabilizing the end of the helical region and promoting a transition to a coil or loop. Glycine’s role in facilitating these structural transitions is well documented, as it introduces conformational flexibility, which is critical for protein dynamics [26,27].

The simulation results for mesco-2 show that the GRGIGRG motif induces the transition from a single helix to a bent helix structure, which potentially facilitates peptide insertion. Moreover, the flexibility of the rotation between the two helices seems to be equally important. When binding occurs with the both helices in parallel to the membrane, insertion is slowed down since the peptide remains stable on the surface with strong electrostatic interactions, as observed in the simulations of a model that had a pre-formed bent helix structure at the N-terminus. However, when this structure forms post-binding and the helices are not coplanar, insertion is facilitated, as demonstrated by simulations of the model with an initial single α-helix configuration. Furthermore, membrane insertion then seems to result in the microscopic disruption of the membrane, which permeabilizes it to propidium iodide, and this likely underlies the bacterial inactivation but does not result in microscopic alterations such as blebbing, as confirmed by AFM.

The observed behavior in both the experiments and simulations resembles closely that of peptides similar to buforin 2 and its analogs, which are known to translocate across lipid membranes without causing significant membrane permeabilization [28,29]. Simulations of a single buforin 2 peptide reveal that it inserts into the membrane without fully translocating or causing significant permeabilization, much like model1. However, with the proper hydration of the membrane, when multiple buforin 2 peptides are present, pore formation occurs [30]. This raises the possibility that pore formation, as observed experimentally in mesco-2, might also occur in simulations under conditions involving multiple peptides.

Therefore, further investigation might involve exploring the mechanism of action of mesco-2. Moreover, the roles of the unstructured C-terminal region and the disulfide bond remain unclear. The results suggest that this region may have lower membrane-binding affinity compared to the N-terminal region. The role of an analogous region at the N-termini of helical anuran AMPs is controversial, and its role in membrane pore formation has not been extensively explored. However, a QSAR study suggests that it could contribute to the peptide’s mode of action [15].

## 4. Materials and Methods

### 4.1. Peptide Identification

The reference genome assembly of *M. corti* (GenBank ID: GCA_900604375.1) and its associated gene models were downloaded from WormBase [31]. The protein translations generated from this annotation track were screened with an in-house script developed at the Department of Life Sciences of the University of Trieste, using a de novo approach to identify candidate sequences displaying features compatible with AMP activity, as previously carried out to successfully identify myticalins in *Mytilus galloprovincialis* [32]. Briefly, only proteins with a length compatible with the usual size of an AMP precursor (i.e., <200 amino acids), which also included a highly supported signal peptide for secretion (detected with SignalP v.6.0) [33] and lacked transmembrane domains (detected with TMHMM v.2.0 [34]), were selected for further analysis. Candidate peptides were further filtered based on a sliding window analysis that took into account 20-residue-long stretches to identify those displaying high cationicity (pI > 10). The resulting hits were then manually inspected, discarding as false positives all sequences (i) showing high homology with proteins previously characterized as not having antimicrobial function (determined via BLASTp searches against UniprotKB) or (ii) bearing conserved domains with known non-antimicrobial function (determined via InteroProScan v.5). The correct annotation of the predicted cDNA and encoded protein sequences of candidate AMPs was checked thanks to the alignment of the available RNA-seq datasets for this species and the reference genome. This task was carried out with the large gap read mapping tool included in the CLC Genomics Workbench v.24 (Qiagen, Hilden, Germany). Briefly, data from the following BioProjects were retrieved: PRJNA433559 [35], PRJEB2679, PRJNA950029 and PRJNA1039817 [36]. The data obtained from the study by Basika and colleagues were further used to assess the gene expression levels for the candidate AMPs in two different developmental stages, i.e., the larval (TT) and strobilated worm (ST) stages. Gene expression analysis was carried out by mapping the reads against the reference genome (length and similarity fraction parameters were set to 0.75 and 0.95, respectively) and computing the gene expression levels as transcripts per million (TPM) [37].

### 4.2. Peptide Synthesis

Mesco-2 was obtained from GenicBio (Shanghai, China) at >95% purity (RP-HPLC using a SinoChrom ODS-BP column, C18, 5 μm, 120 Å, 4.6 mm × 250 mm) (see Appendix A). Chromatographic separation was achieved using a 25−50% acetonitrile/0.1% TFA gradient in 25 min at a 1 mL/min flow rate, and the sequence was confirmed with mass spectrometry. Peptide stock solutions were prepared by dissolving accurately weighed aliquots of the peptide in doubly distilled water, and the concentration was further verified by using the extinction coefficients at 214 nm, calculated as described by Kuipers and Gruppen [38].

### 4.3. Bacterial Strains and Antibacterial Assays

The in vitro antimicrobial testing of the mesco-2 peptide was carried out on Gram-negative laboratory strains obtained from the American Type Culture Collection (ATCC, Rockville, MD, USA), including *E. coli* ATCC 25922, *K. pneumoniae* ATCC 13883, *A. baumannii* ATCC 19606 and *P. aeruginosa* ATCC 27853. Two different strains of *S. aureus*, ATCC 29213 and ATCC 25923, were used as representatives of Gram-positive bacteria. The minimal inhibitory concentrations were determined using the serial two-fold microdilution method, following the recommendations of the European Committee on Antimicrobial Susceptibility Testing (EUCAST) [39]. Bacteria were cultured in fresh Mueller–Hinton broth (MHB) to the mid-log phase and then added to serial dilutions of mesco-2 for a final bacterial load of 5 × 10^5^ CFU/mL of cells in 100 μL per well. The bacteria were then incubated at 37 °C for 18–20 h, and the MIC values were visually determined as the lowest consensus concentration value of the peptide showing no detectable bacterial growth, for an experiment performed at least twice in triplicate.

For the determination of the minimal bactericidal concentration, 5 μL bacterial suspensions from the wells corresponding to MIC, 2× MIC and 4× MIC were plated on MH agar plates. The plates were incubated for 18 h at 37 °C to allow viable colony counting, and the MBC was determined as the peptide killing ≥99.9% of the initial inoculum.

The effect of the mesco-2 peptide on *E. coli* ATCC 25922 and *S. aureus* ATCC 29213 was also assessed in growth kinetics assays. The bacterial growth curves were obtained using mid-log-phase bacteria at 1 × 10^6^ CFU/mL in MHB for a total volume of 200 μL per well, in the presence of increasing peptide concentrations. The optical density (OD) was monitored at 600 nm every 15 min at 37 °C for 12 h in a microplate reader with intermittent shaking (Tecan Infinite Pro 200, Männedorf, Switzerland). Experiments were carried out in triplicate, and the results were expressed as the mean ± S.D.

### 4.4. Membrane Integrity Assay

The integrity of the bacterial cell membrane after exposure to the mesco-2 peptide was assessed as the percentage of propidium iodide (PI)-positive cells, using an Attune NxT flow cytometer (Thermo Fischer Scientific, Waltham, MA, USA), as described previously [40]. *E. coli* ATCC 25922 was incubated to the mid-log phase and PI was added to the bacterial suspension (1 × 10^6^ CFU/mL) at a final concentration of 15 μM (10 μg/mL). Prior to the beginning of the measurement, mesco-2 was added at a concentration corresponding to the MIC (0.5 μM), and measurements were taken at 15, 30 and 90 min. Bacterial cells incubated in MH broth without the peptide were used as negative controls. The experiment was repeated three times, and, for each sample, 20,000 events were acquired. The analysis was carried out with the FCS De Novo Software V7 (De Novo Software, Los Angeles, CA, USA).

### 4.5. Cytotoxicity Assays

The cytotoxic effect of the peptide was determined on a human MEC-1 lymphoid tumor cell line [41]. Cells were cultured in RPMI 1640 medium supplemented with 2 mM L-glutamine, 100 U/mL penicillin, 100 μg/mL streptomycin and 10% (*v*/*v*) fetal bovine serum (FBS) (complete medium) and sub-cultured two–three times a week for a maximum of 20 passages. Cells were counted by the Trypan Blue exclusion test, diluted to 10^6^ cells/mL and plated into 96-well culture plates dispensing 10^5^ cells/well. Metabolic activity was determined after treatment with the peptide in complete medium for 24 h. For this purpose, the colorimetric 3-(4,5-dimethylthiazol2-yl)-2,5-diphenyl tetrazolium bromide (MTT) assay was performed, adding 20 μL of MTT dye (5 mg/mL in PBS) to each well and incubating them for 4 h at 37 °C in the dark. Formazan crystals were solubilized with acidic isopropanol (0.04 N HCl in absolute isopropanol) and the absorbance measured at 570 nm in a microplate reader (Tecan Sunrise, Männedorf, Switzerland). All measurements were carried out at least in triplicate, and the untreated controls were also included.

The effect of the mesco-2 peptide on host cells was further elucidated by a PI uptake assay, using an Attune NxT flow cytometer (Thermo Fischer Scientific, Waltham, MA, USA). For this purpose, MEC-1 cells were removed from the flask and centrifuged and the pellet was resuspended in PBS buffer to a final concentration of 10^6^ cells/mL. PI was then added to the cell suspension for a 15 μM (10 μg/mL) concentration, the suspension was aliquoted into different tubes, and the mesco-2 peptide was added to each tube at increasing concentrations (0.01, 0.1, 1, 10 and 100 μM). The suspension was incubated at 37 °C and the measurements were taken at 15, 30 and 90 min. For each sample, 20,000 events were acquired, keeping the flow rate at 100 µL/min to avoid the formation of clots. The experiment was repeated three times, and the analysis was carried out with the FCS De Novo Software V7 (De Novo Software, Los Angeles, CA, USA).

The experimental data were subjected to computer-assisted statistical analysis, with an ANOVA test and Student–Newman–Keuls post-test, *p* < 0.001 vs. controls and concentrations. Sigmoidal curves were fitted by SigmaPlot, Rsqr 0.99.

### 4.6. Atomic Force Microscopy and Fluorescence Imaging

All AFM measurements were performed in Mueller–Hinton broth at 25 °C and the samples were prepared as follows. To immobilize live bacterial cells for imaging, Petri dishes (WPI, Sarasota, FL, USA) coated with a Cell-Tak (Corning, Corning, NY, USA) solution were prepared as previously reported [19]. An overnight culture of *E. coli* DH5α cells was diluted in fresh Mueller–Hinton broth and propagated for approximately one hour. Subsequently, 50 µL of the prepared culture was incubated in the coated Petri dish for about 20 min. The unbound cells were thoroughly rinsed away with the growth medium, and the immobilized cells were carefully examined under a light microscope to ensure elongation and cell division. After the viability of the immobilized cells was observed, the dish was rinsed again with culture medium and the untreated cells were measured or treated with the peptide at a concentration corresponding to 4 × MIC. All AFM measurements were performed with the Nano-wizard IV system (Bruker, Billerica, MA, USA) in quantitative imaging (QI) mode using the MLCT-BIO-DC (E) probe (Bruker, Billerica, MA, USA). Data were acquired at a 1 µN setpoint with an extension/retraction speed of up to 130 μm/s, while the Z-length was up to 3000 nm at a resolution of 128 × 128 pixels. The collected AFM data were plane- and line-fitted and low-pass-filtered using the JPK v6.4.22 data processing software.

To obtain images of fluorescently stained cells after approximately 3 h of treatment, the culture medium was replaced with sterile physiological saline. Treated cells were stained with the nucleic fluorophores from the LIVE/DEAD™ BacLight™ Bacterial Viability Kit (Thermo Fischer Scientific, Waltham, MA, USA), according to the manufacturer’s instructions. Fluorescence images were taken half an hour after staining, using the IX73 optical fluorescence inverse microscope (Olympus, Tokyo, Japan).

### 4.7. Liposome Preparation

Anionic and neutral small unilamellar vesicles (SUVs) were prepared by dissolving PG:dPG (L-alpha phosphatidylglycerol, diphosphatidylglycerol; 95:5, *w*/*w*) and PC:SM:Ch (L-alpha phosphatidylcholine, sphingomyelin, cholesterol; 40:40:20, *w*/*w/w*), respectively, in CHCl_3_. The solvent was removed by evaporation with a stream of nitrogen to produce a lipid layer and lyophilized overnight. The dry samples were rehydrated with phosphate buffer (10 mM, pH 7.4) for 1 h and sonicated for 45 min. The resulting multilamellar vesicle suspensions were disrupted by several freeze–thaw cycles before passing through a mini-extruder (Croda, Avanti Polar Lipids, Alabaster, AL, USA) successively using polycarbonate filters with 1 μm, 0.4 μm and 0.1 μm pores. The dimensions of the vesicles were verified by dynamic light scattering (Zetasizer Nano, Malvern Panalytical, Malvern, UK), and SUVs were used for the titration of the peptide within 24 h.

### 4.8. Circular Dichroism

CD spectra were obtained on a J-710 spectropolarimeter (Jasco, Tokyo, Japan). The spectra were accumulations of three scans measured in (a) 10 mM sodium phosphate buffer (SPB) solution, (b) different proportions of TFE in SPB, (c) the presence of sodium dodecyl sulphate micelles (10 mM SDS in 10 mM SPB), (d) the presence of anionic SUVs (0.02, 0.04, 0.08 and 0.1 mM) in SPB or (e) the presence of neutral SUVs (0.02, 0.04, 0.08 and 0.1 mM) in SPB. The percentage of helical content was determined as [θ]^222^/[θ]^α^, where [θ]^222^ is the measured molar per residue ellipticity at 222 nm under any given condition and [θ]^α^ is the molar ellipticity for a perfectly formed alpha helix of the same length, estimated as previously described by Chen et al. [42].

### 4.9. Structure Prediction and Molecular Dynamics Simulation Details

Comprehensive analyses were conducted using various prediction servers to evaluate the multifunctional potential of the antimicrobial peptide mesco-2 (see Appendix A).

Structure prediction was performed using two platforms based on the AlphaFold2 software [43]: the AlphaFold Protein Structure Database [44] and ColabFold [45]. AlphaFold DB (https://alphafold.ebi.ac.uk/ accessed on 30 June 2024) [43,44] includes approximately 214 million predicted protein structures derived from all publicly available sequence databases and always conducts searches using full-length proteins. As expected, it recognized mesco-2 as a portion of an uncharacterized protein from *M. corti* (141 amino acids). Consequently, there was a possibility that the resulting 3D model was influenced by the presence of portions of the sequence, such as the signal peptide, which are removed in the cellular environment before protein folding actually starts. Structure prediction was also performed using ColabFold v1.5.5 (https://colab.research.google.com/github/sokrypton/ColabFold/blob/main/AlphaFold2.ipynb accessed on 30 June 2024) [45]. For each prediction, the sequence of mesco-2 specified in Table 1 was used. The main requirement in the predicted models was the inclusion of a disulfide bond.

Molecular dynamics (MD) simulations were performed using the Gromacs 2024.1 package [46]. The CHARMM-GUI Solution Builder [47,48] was used to generate initial conformations of single peptides solvated in water, and the CHARMM-GUI Membrane Builder [49,50] was used to generate initial conformations of solvated peptides in the vicinity of a membrane model. The all-atom (AA) CHARMM36m [51] force field was used, as well as the TIP3 [52] water molecule model. The model membrane was built as a negatively charged bilayer consisting of palmitoyl oleoyl PE (POPE) and palmitoyl oleoyl PG (POPG) in the proportions of 3:1 [53]—specifically, 96 POPE and 32 POPG molecules per layer. Lipid models were obtained from the CHARMM-GUI Individual Lipid Molecule Library [54]. Single solvated peptides were placed on an x–y plane perpendicular to the membrane surface, about 2 nm above the membrane. A layer of water molecules, with a minimum of 4 nm thickness, was added above and below the peptide–membrane system, resulting in about 100 molecules of water per lipid.

The solvated peptide systems were equilibrated in a two-step procedure, while the peptide–membrane systems were processed as per the CHARMM-GUI Membrane Builder recommendation in a six-step procedure [49,50] to a temperature of 310 K and pressure of 1 bar. In all production runs, NpT ensemble conditions were imposed by a Nose–Hoover thermostat and Parrinello–Rahman barostat, with the constants of 1.0 ps for the temperature and 5.0 ps for the pressure, and with compressibility of 4.5 × 10^–^^5^ bar [55,56]. The Leap-frog integrator was used with a fixed timestep of 2 fs, and the bonds were handled by the LINCS algorithm [57]. The Particle Mesh Ewald (PME) method [58] was used to calculate electrostatic interactions, with the Coulomb cutoff at 1.2 nm. The Van der Waals cutoff was set at 1.2 nm, with a force switch at 1.0 nm. For every simulation experiment, two cases (case1 and case2) were used, which differed from each other in terms of the initial velocities. The total simulation time was 500 ns for each simulation.

The post-simulation analysis was performed using Gromacs tools: mindist, density and traj [46]. The DSSP program v3.1.4 [59] was used to generate the secondary structure over the simulation time. Charts were created using Gnuplot 5.4 [60], and the visualization of the systems was achieved using the VMD program v1.9.4 [61].

## 5. Conclusions

We have identified and then synthesized the mesco-2 antimicrobial peptide originating from a parasitic worm, a source still underrepresented in AMP databases. We have experimentally confirmed its potent antibacterial activity against both Gram-positive and Gram-negative bacteria, accompanied by limited toxicity. Additionally, we have confirmed that the mechanism of action of mesco-2 is membrane-related, and, through both experiments and modeling, we have identified the key structural elements that contribute to its specific molecular mechanism of action. Our findings support the potential of mesco-2 for the development of antibacterial agents that could be used to prevent infections.

## Figures and Tables

**Figure 1 ijms-25-11690-f001:**
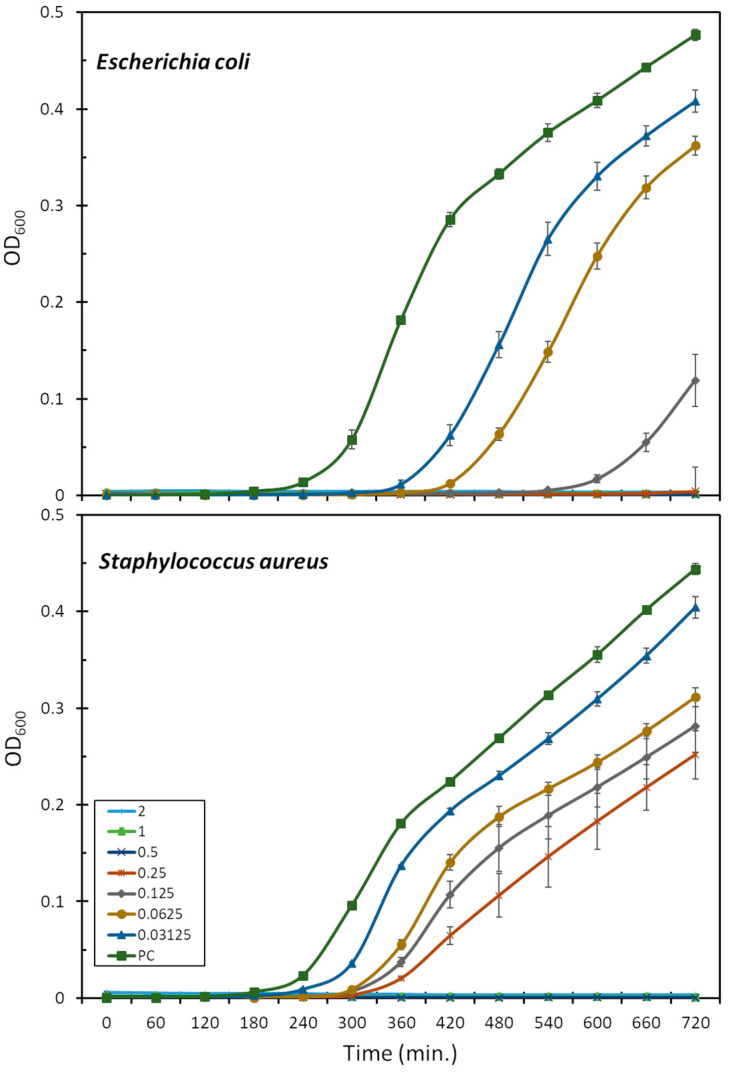
Effect of mesco-2 on bacterial growth kinetics. Growth curves for *E. coli* ATCC 25922 or *S. aureus* ATCC 29213 are shown after incubation with no peptide and mesco-2 at increasing concentrations (0.03–2 μM). Results were obtained by measuring the absorbance at 600 nm for bacteria grown in full Mueller–Hinton broth. PC—positive control.

**Figure 2 ijms-25-11690-f002:**
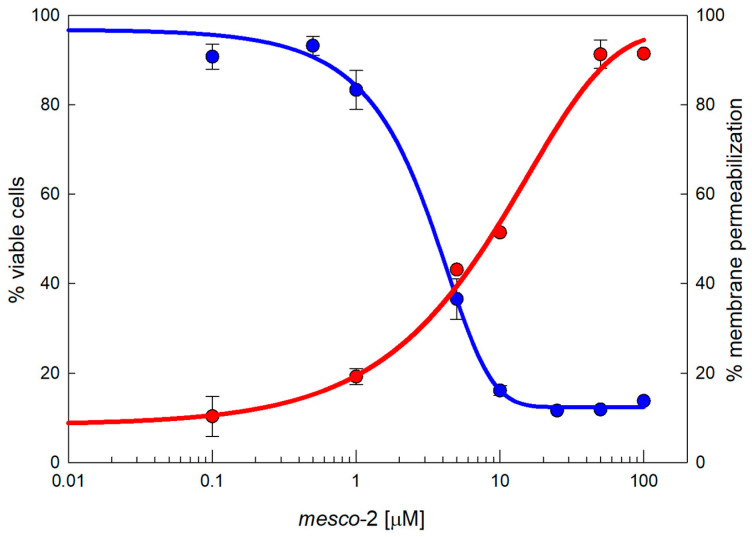
Cytotoxicity of MEC-1 cells exposed to mesco-2. MTT assay (blue line) shows the concentration-dependent effects on the viability of MEC-1 cells (10^6^/mL) treated with the peptide in the growth medium for 24 h; this is overlayed by a red line showing the PI permeabilization of MEC-1 cells treated with mesco-2 for 30 min in PBS buffer. The results are expressed as the mean of three different experiments ± SEM. Sigmoidal curves were fitted by SigmaPlot v14.5.

**Figure 3 ijms-25-11690-f003:**
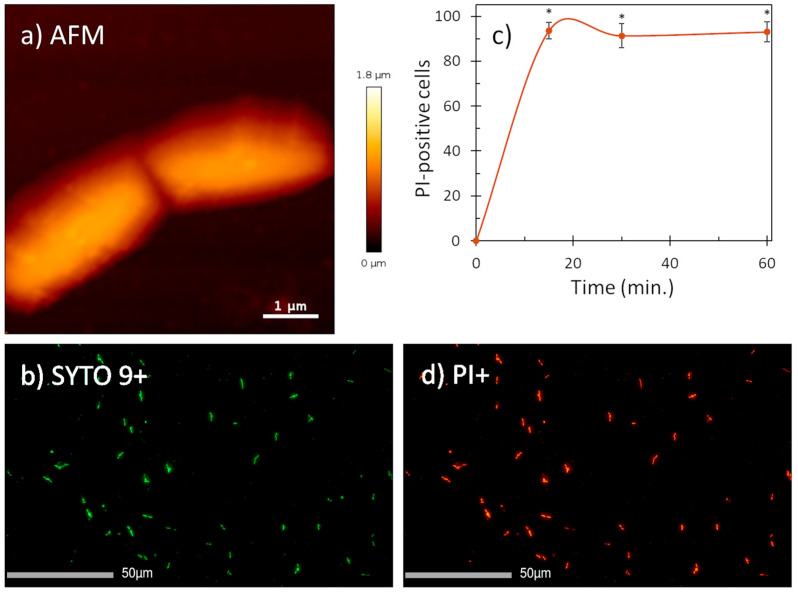
AFM (**a**) and fluorescence images (**b**,**d**) of *E. coli* DH5α cells treated with mesco-2 at 4× MIC. AFM data suggest no visible membrane damage. Fluorescence images (PI+, stained in red; SYTO9+, stained in green) confirm membrane disruption. (**c**) Evaluation of the effect of mesco-2 on bacterial membrane integrity with flow cytometry. Peptide was incubated with *E. coli* ATCC 25922 (1 × 10^6^ CFU/mL) for 60 min at 0.5 μM (MIC). An increase in the signal indicates membrane disruption. Data are expressed as the mean % PI-positive cells ± S.E.M. of three independent experiments. * *p* ˂ 0.05 (*t*-test).

**Figure 4 ijms-25-11690-f004:**
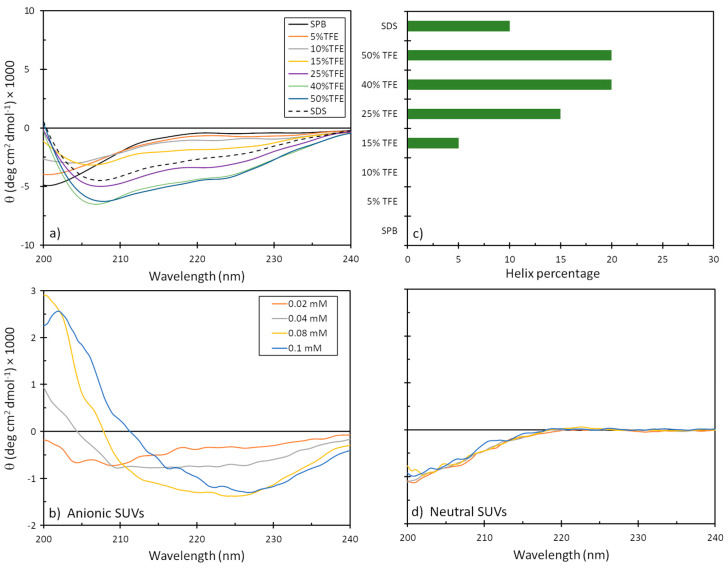
CD spectra of mesco-2 peptide under different conditions. Spectra are the accumulation of three scans carried out with 20 μM peptide in (**a**) SPB, 10 mM SDS in SPB, with different proportions of TFE, and (**b**) anionic and (**d**) neutral SUVs in SPB ((PC:SM:Ch, (L-alpha phosphatidylcholine, sphingomyelin, cholesterol; 40:40:20, *w*/*w*/*w*), (PG:dPG, (L-alpha phosphatidylglycerol, diphosphatidylglycerol; 95:5, *w*/*w*)). (**c**) Precentege of helical content depending on the environment.

**Figure 5 ijms-25-11690-f005:**
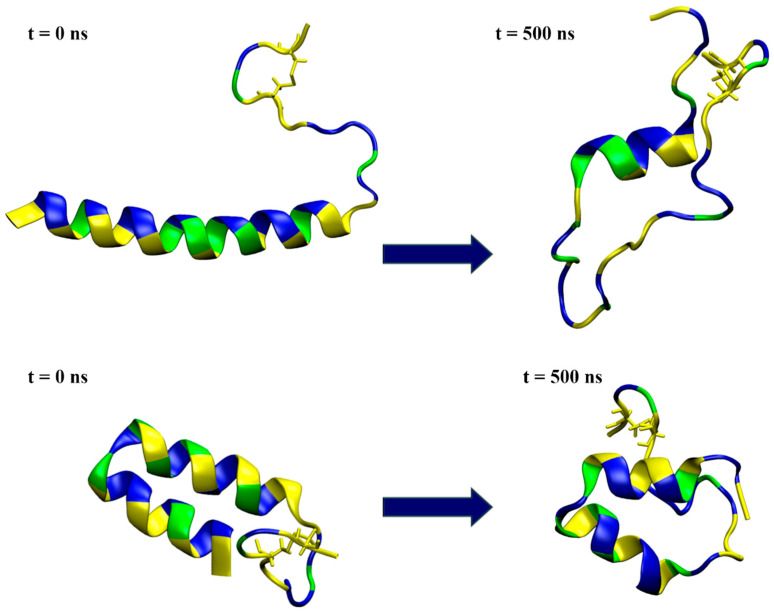
Snapshots from the simulation of a single peptide in water, with the initial conformation depicted on the left and the one at 500 ns simulation time on the right. The top row represents the results for *model1*, while the bottom row shows the *model2* simulations. The peptides are shown in a ribbon representation, with polar residues colored in blue, hydrophobic residues in yellow and glycine in green. Water molecules and ions are excluded for clarity.

**Figure 6 ijms-25-11690-f006:**
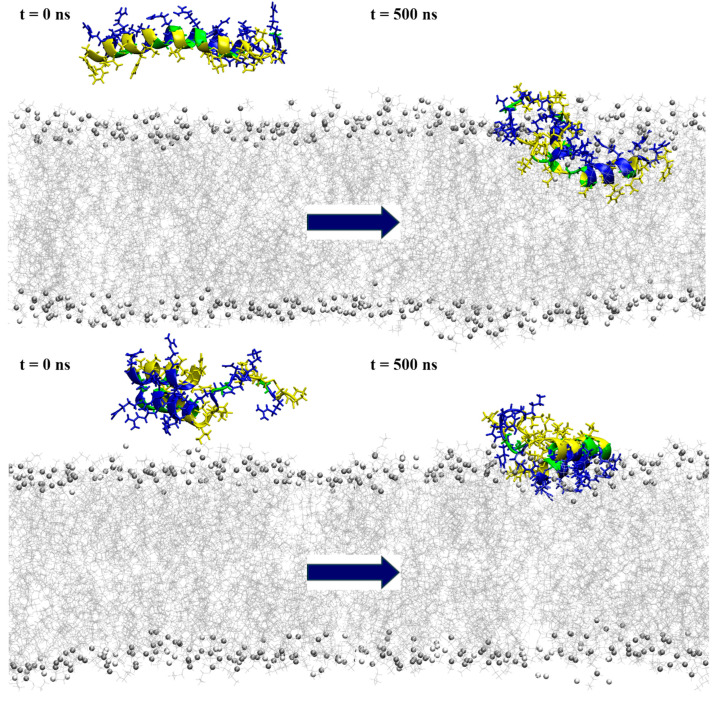
Snapshots from the simulation of a single peptide near the POPE:POPG membrane model, with the initial conformation depicted on the left and the one at 500 ns simulation time on the right. The top row represents the results for *model1*, while the bottom row shows the *model2* simulations. The peptides are shown in a ribbon representation, with polar residues colored in blue, hydrophobic residues in yellow and glycine in green. Lipids are represented in grey (POPE) and white (POPG), with phosphorus atoms depicted as beads and acyl chains as lines. Water molecules and ions are excluded for clarity.

**Figure 7 ijms-25-11690-f007:**
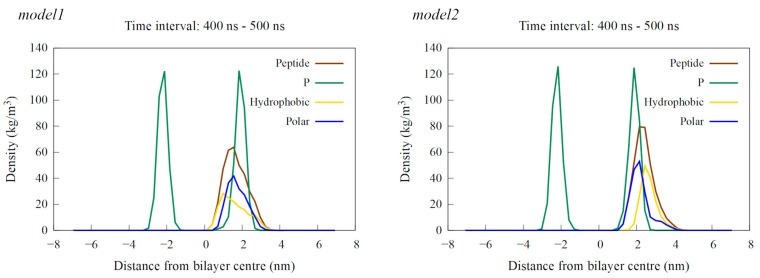
Density profiles calculated as an average over the last 100 ns of simulation time for a single peptide near the POPE:POPG membrane. The results for the *model1* simulation are shown on the left, and those for the *model2* simulation are on the right. The profiles show the density of the entire peptide in brown, hydrophobic residues in yellow, polar residues in blue and phosphorus (P) atoms, representing the membrane’s polar region, in green.

**Table 1 ijms-25-11690-t001:** Sequences and physicochemical properties of mesco peptides.

Name	Sequence	Charge	H ^1^	H ^rel 2^
mesco-1	WRRLRRRISGGLRRIFRKPRRICFPYCPTGPRYPGPRPY	+13	−0.475	0.071
mesco-2	FFRRIGRAFSRVGRGIGRGFRQLGRLMPRGNYKICLGRCP	+11	−0.232	0.106
mesco-3	FLRRIGRAFSRVGRGIGRGFRQLGRLMPRGNYRICLGRCPR	+12	−0.316	0.095

^1^ Hydrophobicity, calculated using the CCS scale [16]. ^2^ Hydrophobic moment relative to a perfectly amphipathic helical peptide of 18 residues.

**Table 2 ijms-25-11690-t002:** Antimicrobial activity of mesco-2 in MHB, expressed as MIC and MBC (μM).

Bacterial Strain	Mesco-2	
	MIC	MBC
*Escherichia coli* ATCC 25922	0.5	0.5
*Acinetobacter baumannii* ATCC 19606	1–2	1–2
*Klebsiella pneumoniae* ATCC 13883	0.5	0.5
*Pseudomonas aeruginosa* ATCC 27853	2	2
*Staphylococcus aureus* ATCC 29213	1	1
*Staphylococcus aureus* ATCC 25923	2	8

## Data Availability

The data presented in this study are available on request from the corresponding author.

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
