# Peer review of "Antimicrobial Peptide with a Bent Helix Motif Identified in Parasitic Flatworm Mesocestoides corti"

_ijms, 2024, doi:10.3390/ijms252111690_

Round 1

Reviewer 1 Report

Comments and Suggestions for Authors

The manuscript focuses on experimental and computational analysis of an antimicrobial peptide, mesco-2.

Overall the results are convincing and consistent.  The suggestions below are primarily to improve the presentation and connection to existing literature, but there is one experimental question:

1. In figure 1, the Y-axis should be OD600.  Light scattering is technically not equivalent to Absorbance, although routinely performed in absorbance instruments.  I would also strongly recommend adding symbols at the data points for greater clarity

2. In figure 1, the PC term in the legend is not described.

3. In figure 3C, were controls performed with untreated cells?  This is an important control to mention or show, to compare spontaneous leakage/crossing of PI compared to the peptide-treated samples

4.  In figure 4, there are no panel labels (A, B, C, D).  

5. In figure 4, i would recommend a different color line for SDS in the upper left panel.  It is difficult to distinguish from the others.

6. In the interpretation of helical content, why is the SDS sample so much lower than 40 and 50% TFE?  These spectra appear to be nearly superimposable.

7. The authors should include more in the discussion about the helical conformation shown to more deeply insert in the membrane (ColabFold).  There are numerous examples in literature about penetration depth of peptides into bilayers and correlations with membrane permeabilization.  

8.  The authors should add a paragraph discussing the role of Arg in this peptide.  Peptides with multi-Arg residues are well known to promote transmembrane transport, without causing significant large scale disruptions to the bilayer structure (such as in the TAT sequence).  This is consistent with the authors' data showing PI permeabilization without structural disruption (AFM data).  

9. The authors should mention the similarity of model1 with the well studied AMP LL-37.  Structurally, by eye, these helices seem very similar.  While model1 is just that, a computational model, any structural parallels to existing, experimentally determined structures would be beneficial.

10. the authors should give a general proofreading.  For examples: (A) line 218 should be Cys36 and Cys39, not Cis.

Reviewer 2 Report

Comments and Suggestions for Authors
  1. Line 18Mesocestoides corti should be italicized to follow standard conventions for species names.
  2. Line 42: please add references for this sentence “Antimicrobial peptides (AMPs) are endogenous molecules that are synthesized as part of the innate defenses of all multicellular organisms and have the potential to mitigate antimicrobial resistance.” (for example doi: 10.3389/fcimb.2021.668632)
  3. Line 51in vivo should be italicized as it is a Latin phrase commonly italicized in academic writing.
  4. Line 45: The use of "and/or" is considered informal for a scientific article. A better alternative would be to rephrase it as: "and may also have immunomodulatory effects."
  5. Line 294: please add a reference for this sentence “A variety of online servers predicted that mesco-2 has high potential for antimicrobial activity, along with low toxicity, supporting its functionality as an antimicrobial peptide” (for example DOI: 10.2174/1381612828666220817163339)
  6. Avoid first-person narration: In scientific writing, it is generally preferable to use the third person. For example, instead of "We observed that," you could say "It was observed that" or "The results showed that."
  7. Justify the use of a tumor cell line for cytotoxicity evaluation: The explanation for choosing a tumor cell line to assess peptide cytotoxicity should be more detailed. The rationale for the use of cells in suspension as an alternative to adherent cells is not clearly explained.
